# Learning Mean-Field Games

**Xin Guo**
University of California, Berkeley
`xinguo@berkeley.edu`

**Anran Hu**
University of California, Berkeley
`anran_hu@berkeley.edu`

**Renyuan Xu**
University of California, Berkeley
`renyuanxu@berkeley.edu`

**Junzi Zhang**
Stanford University
`junziz@stanford.edu`

## Abstract

This paper presents a general mean-field game (GMFG) framework for simultaneous learning and decision-making in stochastic games with a large population. It first establishes the existence of a unique Nash Equilibrium to this GMFG, and explains that naively combining Q-learning with the fixed-point approach in classical MFGs yields unstable algorithms. It then proposes a Q-learning algorithm with Boltzmann policy (GMF-Q), with analysis of convergence property and computational complexity. The experiments on repeated Ad auction problems demonstrate that this GMF-Q algorithm is efficient and robust in terms of convergence and learning accuracy. Moreover, its performance is superior in convergence, stability, and learning ability, when compared with existing algorithms for multi-agent reinforcement learning.

## 1   Introduction

**Motivating example.**   This paper is motivated by the following Ad auction problem for an advertiser. An Ad auction is a stochastic game on an Ad exchange platform among a large number of players, the advertisers. In between the time a web user requests a page and the time the page is displayed, usually within a millisecond, a Vickrey-type of second-best-price auction is run to incentivize interested advertisers to bid for an Ad slot to display advertisement. Each advertiser has limited information before each bid: first, her own *valuation* for a slot depends on an unknown conversion of clicks for the item; secondly, she, should she win the bid, only knows the reward *after* the user's activities on the website are finished. In addition, she has a budget constraint in this repeated auction.

The question is, how should she bid in this online sequential repeated game when there is a *large* population of bidders competing on the Ad platform, with *unknown* distributions of the conversion of clicks and rewards?

Besides the Ad auction, there are many real-world problems involving a large number of players and unknown systems. Examples include massive multi-player online role-playing games [19], high frequency tradings [24], and the sharing economy [13].

**Our work.**   Motivated by these problems, we consider a general framework of simultaneous learning and decision-making in stochastic games with a large population. We formulate a general mean-field-game (GMFG) with incorporation of action distributions, (randomized) relaxed policies, and with unknown rewards and dynamics. This general framework can also be viewed as a generalized version of MFGs of McKean-Vlasov type [1], which is a different paradigm from the classical MFG. It is also beyond the scope of the existing Q-learning framework for Markov decision problem (MDP) with unknown distributions, as MDP is technically equivalent to a single player stochastic game.

On the theory front, this general framework differs from all existing MFGs. We establish under appropriate technical conditions, the existence and uniqueness of the Nash equilibrium (NE) to this GMFG. On the computational front, we show that naively combining Q-learning with the three-step fixed-point approach in classical MFGs yields unstable algorithms. We then propose a Q-learning algorithm with Boltzmann policy (GMF-Q), establish its convergence property and analyze its computational complexity. Finally, we apply this GMF-Q algorithm to the Ad auction problem, where this GMF-Q algorithm demonstrates its efficiency and robustness in terms of convergence and learning. Moreover, its performance is superior, when compared with existing algorithms for multi-agent reinforcement learning for convergence, stability, and learning accuracy.

**Related works.** On learning large population games with mean-field approximations, [39] focuses on inverse reinforcement learning for MFGs without decision making, [40] studies an MARL problem with a first-order mean-field approximation term modeling the interaction between one player and all the other finite players, and [22] and [41] consider model-based adaptive learning for MFGs in specific models (*e.g.*, linear-quadratic and oscillator games). More recently, [26] studies the local convergence of actor-critic algorithms on finite time horizon MFGs, and [34] proposes a policy-gradient based algorithm and analyzes the so-called local NE for reinforcement learning in infinite time horizon MFGs. For learning large population games without mean-field approximation, see [14, 21] and the references therein. In the specific topic of learning auctions with a large number of advertisers, [6] and [20] explore reinforcement learning techniques to search for social optimal solutions with real-word data, and [18] uses MFGs to model the auction system with unknown conversion of clicks within a Bayesian framework.

However, none of these works consider the problem of simultaneous learning and decision-making in a general MFG framework. Neither do they establish the existence and uniqueness of the (global) NE, nor do they present model-free learning algorithms with complexity analysis and convergence to the NE. Note that in principle, global results are harder to obtain compared to local results.

## 2 Framework of General MFG (GMFG)

### 2.1 Background: classical $N$-player Markovian game and MFG

Let us first recall the classical $N$-player game. There are $N$ players in a game. At each step $t$, the state of player $i$ $(= 1, 2, \cdots, N)$ is $s_t^i \in \mathcal{S} \subseteq \mathbb{R}^d$ and she takes an action $a_t^i \in \mathcal{A} \subseteq \mathbb{R}^p$. Here $d, p$ are positive integers, and $\mathcal{S}$ and $\mathcal{A}$ are compact (for example, finite) state space and action space, respectively. Given the current state profile of $N$-players $\mathbf{s}_t = (s_t^1, \ldots, s_t^N) \in \mathcal{S}^N$ and the action $a_t^i$, player $i$ will receive a reward $r^i(\mathbf{s}_t, a_t^i)$ and her state will change to $s_{t+1}^i$ according to a transition probability function $P^i(\mathbf{s}_t, a_t^i)$.

A Markovian game further restricts the admissible policy/control for player $i$ to be of the form $a_t^i = \pi_t^i(\mathbf{s}_t)$. That is, $\pi_t^i : \mathcal{S}^N \to \mathcal{P}(\mathcal{A})$ maps each state profile $\mathbf{s} \in \mathcal{S}^N$ to a randomized action, with $\mathcal{P}(\mathcal{X})$ the space of probability measures on space $\mathcal{X}$. The accumulated reward (a.k.a. the value function) for player $i$, given the initial state profile $\mathbf{s}$ and the policy profile sequence $\boldsymbol{\pi} := \{\boldsymbol{\pi}_t\}_{t=0}^{\infty}$ with $\boldsymbol{\pi}_t = (\pi_t^1, \ldots, \pi_t^N)$, is then defined as

$$V^i(\mathbf{s}, \boldsymbol{\pi}) := \mathbb{E}\left[\sum_{t=0}^{\infty} \gamma^t r^i(\mathbf{s}_t, a_t^i)\Big|\mathbf{s}_0 = \mathbf{s}\right], \tag{1}$$

where $\gamma \in (0, 1)$ is the discount factor, $a_t^i \sim \pi_t^i(s^t)$, and $s_{t+1}^i \sim P^i(\mathbf{s}_t, a_t^i)$. The goal of each player is to maximize her value function over all admissible policy sequences.

In general, this type of stochastic $N$-player game is notoriously hard to analyze, especially when $N$ is large [28]. Mean field game (MFG), pioneered by [17] and [23] in the continuous settings and later developed in [4, 10, 16, 25, 33] for discrete settings, provides an ingenious and tractable aggregation approach to approximate the otherwise challenging $N$-player stochastic games. The basic idea for an MFG goes as follows. Assume all players are identical, indistinguishable and interchangeable, when $N \to \infty$, one can view the limit of other players' states $\mathbf{s}_t^{-i} = (s_t^1, \ldots, s_t^{i-1}, s_t^{i+1}, \ldots, s_t^N)$ as a population state distribution $\mu_t$ with $\mu_t(s) := \lim_{N \to \infty} \frac{\sum_{j=1, j \neq i}^N \mathbf{I}_{s_t^j = s}}{N}$.[1] Due to the homogeneity

of the players, one can then focus on a single (representative) player. That is, in an MFG, one may consider instead the following optimization problem,

$$\text{maximize}_{\boldsymbol{\pi}} \quad V(s, \boldsymbol{\pi}, \boldsymbol{\mu}) := \mathbb{E}\left[\sum_{t=0}^{\infty} \gamma^t r(s_t, a_t, \mu_t)|s_0 = s\right]$$
$$\text{subject to} \quad s_{t+1} \sim P(s_t, a_t, \mu_t), \quad a_t \sim \pi_t(s_t, \mu_t),$$

where $\boldsymbol{\pi} := \{\pi_t\}_{t=0}^{\infty}$ denotes the policy sequence and $\boldsymbol{\mu} := \{\mu_t\}_{t=0}^{\infty}$ the distribution flow. In this MFG setting, at time $t$, after the representative player chooses her action $a_t$ according to some policy $\pi_t$, she will receive reward $r(s_t, a_t, \mu_t)$ and her state will evolve under a *controlled stochastic dynamics* of a mean-field type $P(\cdot|s_t, a_t, \mu_t)$. Here the policy $\pi_t$ depends on both the current state $s_t$ and the current population state distribution $\mu_t$ such that $\pi : \mathcal{S} \times \mathcal{P}(\mathcal{S}) \to \mathcal{P}(\mathcal{A})$.

## 2.2 General MFG (GMFG)

In the classical MFG setting, the reward and the dynamic for each player are known. They depend only on $s_t$ the state of the player, $a_t$ the action of this particular player, and $\mu_t$ the population state distribution. In contrast, in the motivating auction example, the reward and the dynamic are unknown; they rely on the actions of *all* players, as well as on $s_t$ and $\mu_t$.

We therefore define the following general MFG (GMFG) framework. At time $t$, after the representative player chooses her action $a_t$ according to some policy $\pi : \mathcal{S} \times \mathcal{P}(\mathcal{S}) \to \mathcal{P}(\mathcal{A})$, she will receive a reward $r(s_t, a_t, \mathcal{L}_t)$ and her state will evolve according to $P(\cdot|s_t, a_t, \mathcal{L}_t)$, where $r$ and $P$ are possibly unknown. The objective of the player is to solve the following control problem:

$$\text{maximize}_{\boldsymbol{\pi}} \quad V(s, \boldsymbol{\pi}, \boldsymbol{\mathcal{L}}) := \mathbb{E}\left[\sum_{t=0}^{\infty} \gamma^t r(s_t, a_t, \mathcal{L}_t)|s_0 = s\right] \qquad \text{(GMFG)}$$
$$\text{subject to} \quad s_{t+1} \sim P(s_t, a_t, \mathcal{L}_t), \quad a_t \sim \pi_t(s_t, \mu_t).$$

Here, $\boldsymbol{\mathcal{L}} := \{\mathcal{L}_t\}_{t=0}^{\infty}$, with $\mathcal{L}_t = \mathbb{P}_{s_t, a_t} \in \mathcal{P}(\mathcal{S} \times \mathcal{A})$ the joint distribution of the state and the action (*i.e.*, the population state-action pair). $\mathcal{L}_t$ has marginal distributions $\alpha_t$ for the population action and $\mu_t$ for the population state. Notice that $\{\mathcal{L}_t\}_{t=0}^{\infty}$ could depend on time. Namely, an infinite time horizon MFG could still have time-dependent NE solution due to the mean information process (game interaction) in the MFG. This is fundamentally different from the theory of single-agent MDP where the optimal control, if exists uniquely, would be time independent in an infinite time horizon setting.

In this framework, we adopt the well-known Nash Equilibrium (NE) for analyzing stochastic games.

**Definition 2.1** (NE for GMFGs). *In* (GMFG), *a player-population profile* $(\boldsymbol{\pi}^\star, \boldsymbol{\mathcal{L}}^\star) := (\{\pi_t^\star\}_{t=0}^{\infty}, \{\mathcal{L}_t^\star\}_{t=0}^{\infty})$ *is called an NE if*

1. *(Single player side) Fix* $\boldsymbol{\mathcal{L}}^\star$, *for any policy sequence* $\boldsymbol{\pi} := \{\pi_t\}_{t=0}^{\infty}$ *and initial state* $s \in \mathcal{S}$,

$$V(s, \boldsymbol{\pi}^\star, \boldsymbol{\mathcal{L}}^\star) \geq V(s, \boldsymbol{\pi}, \boldsymbol{\mathcal{L}}^\star). \qquad (2)$$

2. *(Population side)* $\mathbb{P}_{s_t, a_t} = \mathcal{L}_t^\star$ *for all* $t \geq 0$, *where* $\{s_t, a_t\}_{t=0}^{\infty}$ *is the dynamics under the policy sequence* $\boldsymbol{\pi}^\star$ *starting from* $s_0 \sim \mu_0^\star$, *with* $a_t \sim \pi_t^\star(s_t, \mu_t^\star)$, $s_{t+1} \sim P(\cdot|s_t, a_t, \mathcal{L}_t^\star)$, *and* $\mu_t^\star$ *being the population state marginal of* $\mathcal{L}_t^\star$.

The single player side condition captures the optimality of $\boldsymbol{\pi}^\star$, when the population side is fixed. The population side condition ensures the "consistency" of the solution: it guarantees that the state and action distribution flow of the single player does match the population state and action sequence $\boldsymbol{\mathcal{L}}^\star$.

## 2.3 Example: GMFG for the repeated auction

Now, consider the repeated Vickrey auction with a budget constraint in Section 1. Take a representative advertiser in the auction. Denote $s_t \in \{0, 1, 2, \cdots, s_{\max}\}$ as the budget of this player at time $t$, where $s_{\max} \in \mathbb{N}^+$ is the maximum budget allowed on the Ad exchange with a unit bidding price. Denote $a_t$ as the bid price submitted by this player and $\alpha_t$ as the bidding/(action) distribution of the population. The reward for this advertiser with bid $a_t$ and budget $s_t$ is

$$r_t = \mathbf{I}_{w_t^M = 1}\left[(v_t - a_t^M) - (1 + \rho)\mathbf{I}_{s_t < a_t^M}(a_t^M - s_t)\right]. \qquad (3)$$

Here $w_t^M$ takes values 1 and 0, with $w_t^M = 1$ meaning this player winning the bid and 0 otherwise. The probability of winning the bid would depend on $M$, the index for the game intensity, and $\alpha_t$. (See discussion on $M$ in Appendix H.1.) The conversion of clicks at time $t$ is $v_t$ and follows an unknown distribution. $a_t^M$ is the value of the second largest bid at time $t$, taking values from 0 to $s_{\max}$, and depends on both $M$ and $\mathcal{L}_t$. Should the player win the bid, the reward $r_t$ consists of two parts, corresponding to the two terms in (3). The first term is the profit of wining the auction, as the winner only needs to pay for the second best bid $a_t^M$ in a Vickrey auction. The second term is the penalty of overshooting if the payment exceeds her budget, with a penalty rate $\rho$. At each time $t$, the budget dynamics $s_t$ follows,

$$
s_{t+1} = \begin{cases} s_t, & w_t^M \neq 1, \\ s_t - a_t^M, & w_t^M = 1 \text{ and } a_t^M \leq s_t, \\ 0, & w_t^M = 1 \text{ and } a_t^M > s_t. \end{cases}
$$

That is, if this player does not win the bid, the budget will remain the same. If she wins and has enough money to pay, her budget will decrease from $s_t$ to $s_t - a_t^M$. However, if she wins but does not have enough money, her budget will be 0 after the payment and there will be a penalty in the reward function. Note that in this game, both the rewards $r_t$ and the dynamics $s_t$ are unknown *a priori*.

In practice, one often modifies the dynamics of $s_{t+1}$ with a non-negative random budget fulfillment $\Delta(s_{t+1})$ after the auction clearing [11], such that $\hat{s}_{t+1} = s_{t+1} + \Delta(s_{t+1})$. One may see some particular choices of $\Delta(s_{t+1})$ in the experiment section (Section 5).

## 3 Solution for GMFGs

We now establish the existence and uniqueness of the NE to (GMFG), by generalizing the classical fixed-point approach for MFGs to this GMFG setting. (See [17] and [23] for the classical case). It consists of three steps.

**Step A.** Fix $\boldsymbol{\mathcal{L}} := \{\mathcal{L}_t\}_{t=0}^{\infty}$, (GMFG) becomes the classical optimization problem. Indeed, with $\boldsymbol{\mathcal{L}}$ fixed, the population state distribution sequence $\boldsymbol{\mu} := \{\mu_t\}_{t=0}^{\infty}$ is also fixed, hence the space of admissible policies is reduced to the single-player case. Solving (GMFG) is now reduced to finding a policy sequence $\pi_{t,\boldsymbol{\mathcal{L}}}^{\star} \in \Pi := \{\pi \mid \pi : \mathcal{S} \to \mathcal{P}(\mathcal{A})\}$ over all admissible $\boldsymbol{\pi}_{\boldsymbol{\mathcal{L}}} = \{\pi_{t,\boldsymbol{\mathcal{L}}}\}_{t=0}^{\infty}$, to maximize

$$
\begin{aligned}
V(s, \boldsymbol{\pi}_{\boldsymbol{\mathcal{L}}}, \boldsymbol{\mathcal{L}}) := & \ \mathbb{E}\left[\sum_{t=0}^{\infty} \gamma^t r(s_t, a_t, \mathcal{L}_t) | s_0 = s\right], \\
\text{subject to} & \quad s_{t+1} \sim P(s_t, a_t, \mathcal{L}_t), \quad a_t \sim \pi_{t,\boldsymbol{\mathcal{L}}}(s_t).
\end{aligned}
$$

Notice that with $\boldsymbol{\mathcal{L}}$ fixed, one can safely suppress the dependency on $\mu_t$ in the admissible policies. Moreover, given this fixed $\boldsymbol{\mathcal{L}}$ sequence and the solution $\boldsymbol{\pi}_{\boldsymbol{\mathcal{L}}}^{\star} := \{\pi_{t,\boldsymbol{\mathcal{L}}}^{\star}\}_{t=0}^{\infty}$, one can define a mapping from the fixed population distribution sequence $\boldsymbol{\mathcal{L}}$ to an arbitrarily chosen optimal randomized policy sequence. That is,

$$
\Gamma_1 : \{\mathcal{P}(\mathcal{S} \times \mathcal{A})\}_{t=0}^{\infty} \to \{\Pi\}_{t=0}^{\infty},
$$

such that $\boldsymbol{\pi}_{\boldsymbol{\mathcal{L}}}^{\star} = \Gamma_1(\boldsymbol{\mathcal{L}})$. Note that this $\boldsymbol{\pi}_{\boldsymbol{\mathcal{L}}}^{\star}$ sequence satisfies the single player side condition in Definition 2.1 for the population state-action pair sequence $\boldsymbol{\mathcal{L}}$. That is, $V(s, \boldsymbol{\pi}_{\boldsymbol{\mathcal{L}}}^{\star}, \boldsymbol{\mathcal{L}}) \geq V(s, \boldsymbol{\pi}, \boldsymbol{\mathcal{L}})$, for any policy sequence $\boldsymbol{\pi} = \{\pi_t\}_{t=0}^{\infty}$ and any initial state $s \in \mathcal{S}$.

As in the MFG literature [17], a feedback regularity condition is needed for analyzing Step A.

**Assumption 1.** *There exists a constant $d_1 \geq 0$, such that for any $\boldsymbol{\mathcal{L}}, \boldsymbol{\mathcal{L}}' \in \{\mathcal{P}(\mathcal{S} \times \mathcal{A})\}_{t=0}^{\infty}$,*

$$
D(\Gamma_1(\boldsymbol{\mathcal{L}}), \Gamma_1(\boldsymbol{\mathcal{L}}')) \leq d_1 \mathcal{W}_1(\boldsymbol{\mathcal{L}}, \boldsymbol{\mathcal{L}}'), \tag{4}
$$

*where*

$$
\begin{aligned}
D(\boldsymbol{\pi}, \boldsymbol{\pi}') &:= \sup_{s \in \mathcal{S}} \mathcal{W}_1(\boldsymbol{\pi}(s), \boldsymbol{\pi}'(s)) = \sup_{s \in \mathcal{S}} \sup_{t \in \mathbb{N}} W_1(\pi_t(s), \pi_t'(s)), \\
\mathcal{W}_1(\boldsymbol{\mathcal{L}}, \boldsymbol{\mathcal{L}}') &:= \sup_{t \in \mathbb{N}} W_1(\mathcal{L}_t, \mathcal{L}_t'),
\end{aligned} \tag{5}
$$

*and $W_1$ is the $\ell_1$-Wasserstein distance between probability measures [9, 31, 37].*

**Step B.** Based on the analysis in Step A and $\boldsymbol{\pi}_{\mathcal{L}}^{\star} = \{\pi_{t,\mathcal{L}}^{\star}\}_{t=0}^{\infty}$, update the initial sequence $\mathcal{L}$ to $\mathcal{L}'$ following the controlled dynamics $P(\cdot|s_t, a_t, \mathcal{L}_t)$.

Accordingly, for any admissible policy sequence $\boldsymbol{\pi} \in \{\Pi\}_{t=0}^{\infty}$ and a joint population state-action pair sequence $\mathcal{L} \in \{\mathcal{P}(\mathcal{S} \times \mathcal{A})\}_{t=0}^{\infty}$, define a mapping $\Gamma_2 : \{\Pi\}_{t=0}^{\infty} \times \{\mathcal{P}(\mathcal{S} \times \mathcal{A})\}_{t=0}^{\infty} \to \{\mathcal{P}(\mathcal{S} \times \mathcal{A})\}_{t=0}^{\infty}$ as follows:

$$\Gamma_2(\boldsymbol{\pi}, \mathcal{L}) := \hat{\mathcal{L}} = \{\mathbb{P}_{s_t, a_t}\}_{t=0}^{\infty}, \tag{6}$$

where $s_{t+1} \sim \mu_t P(\cdot|\cdot, a_t, \mathcal{L}_t)$, $a_t \sim \pi_t(s_t)$, $s_0 \sim \mu_0$, and $\mu_t$ is the population state marginal of $\mathcal{L}_t$.

One needs a standard assumption in this step.

**Assumption 2.** *There exist constants $d_2$, $d_3 \geq 0$, such that for any admissible policy sequences $\boldsymbol{\pi}, \boldsymbol{\pi}^1, \boldsymbol{\pi}^2$ and joint distribution sequences $\mathcal{L}, \mathcal{L}^1, \mathcal{L}^2$,*

$$\mathcal{W}_1(\Gamma_2(\boldsymbol{\pi}^1, \mathcal{L}), \Gamma_2(\boldsymbol{\pi}^2, \mathcal{L})) \leq d_2 D(\boldsymbol{\pi}^1, \boldsymbol{\pi}^2), \tag{7}$$

$$\mathcal{W}_1(\Gamma_2(\boldsymbol{\pi}, \mathcal{L}^1), \Gamma_2(\boldsymbol{\pi}, \mathcal{L}^2)) \leq d_3 \mathcal{W}_1(\mathcal{L}^1, \mathcal{L}^2). \tag{8}$$

Assumption 2 can be reduced to Lipschitz continuity and boundedness of the transition dynamics $P$. (See the Appendix for more details.)

**Step C.** Repeat Step A and Step B until $\mathcal{L}'$ matches $\mathcal{L}$.

This step is to take care of the population side condition. To ensure the convergence of the combined step A and step B, it suffices if $\Gamma : \{\mathcal{P}(\mathcal{S} \times \mathcal{A})\}_{t=0}^{\infty} \to \{\mathcal{P}(\mathcal{S} \times \mathcal{A})\}_{t=0}^{\infty}$ is a contractive mapping under the $\mathcal{W}_1$ distance, with $\Gamma(\mathcal{L}) := \Gamma_2(\Gamma_1(\mathcal{L}), \mathcal{L})$. Then by the Banach fixed point theorem and the completeness of the related metric spaces, there exists a unique NE to the GMFG.

In summary, we have

**Theorem 1** (Existence and Uniqueness of GMFG solution). *Given Assumptions 1 and 2, and assuming that $d_1 d_2 + d_3 < 1$, there exists a unique NE to* (GMFG).

## 4 RL Algorithms for (stationary) GMFGs

In this section, we design the computational algorithm for the GMFG. Since the reward and transition distributions are unknown, this is simultaneously learning the system and finding the NE of the game. We will focus on the case with finite state and action spaces, *i.e.*, $|\mathcal{S}|, |\mathcal{A}| < \infty$. We will look for stationary (time independent) NEs. Accordingly, we abbreviate $\boldsymbol{\pi} := \{\pi\}_{t=0}^{\infty}$ and $\mathcal{L} := \{\mathcal{L}\}_{t=0}^{\infty}$ as $\pi$ and $\mathcal{L}$, respectively. This stationarity property enables developing appropriate time-independent Q-learning algorithm, suitable for an infinite time horizon game. Modification from the GMFG framework to this special stationary setting is straightforward, and is left to Appendix B. Note that the assumptions to guarantee the existence and uniqueness of GMFG solutions are slightly different between the stationary and non-stationary cases. For instance, one can compare (7)-(8) with (21)-(22).

The algorithm consists of two steps, parallel to Step $A$ and Step $B$ in Section 3.

**Step 1: Q-learning with stability for fixed $\mathcal{L}$.** With $\mathcal{L}$ fixed, it becomes a standard learning problem for an infinite horizon MDP. We will focus on the Q-learning algorithm [35, 32].

The Q-learning algorithm approximates the value iteration by stochastic approximation. At each step with the state $s$ and an action $a$, the system reaches state $s'$ according to the controlled dynamics and the Q-function is updated according to

$$Q_{\mathcal{L}}(s, a) \leftarrow (1 - \beta_t(s, a))Q_{\mathcal{L}}(s, a) + \beta_t(s, a) \left[r(s, a, \mathcal{L}) + \gamma \max_{\tilde{a}} Q_{\mathcal{L}}(s', \tilde{a})\right], \tag{9}$$

where the step size $\beta_t(s, a)$ can be chosen as (*cf.* [7])

$$\beta_t(s, a) = \begin{cases} |\#(s, a, t) + 1|^{-h}, & (s, a) = (s_t, a_t), \\ 0, & \text{otherwise.} \end{cases}$$

with $h \in (1/2, 1)$. Here $\#(s, a, t)$ is the number of times up to time $t$ that one visits the pair $(s, a)$. The algorithm then proceeds to choose action $a'$ based on $Q_{\mathcal{L}}$ with appropriate exploration strategies, including the $\epsilon$-greedy strategy.

After obtaining the approximate $\hat{Q}_{\mathcal{L}}^{\star}$, in order to retrieve an approximately optimal policy, it would be natural to define an **argmax-e** operator so that actions with equal maximum Q-values would have equal probabilities to be selected. Unfortunately, the discontinuity and sensitivity of **argmax-e** could lead to an unstable algorithm (see Figure 4 for the corresponding naive Algorithm 2 in Appendix). [2]

Instead, we consider a Boltzmann policy based on the operator $\textbf{softmax}_c : \mathbb{R}^n \to \mathbb{R}^n$, defined as

$$\textbf{softmax}_c(x)_i = \frac{\exp(cx_i)}{\sum_{j=1}^n \exp(cx_j)}. \tag{10}$$

This operator is smooth and close to the **argmax-e** (see Lemma 7 in the Appendix). Moreover, even though Boltzmann policies are not optimal, the difference between the Boltzmann and the optimal one can always be controlled by choosing the hyper-parameter $c$ appropriately in the **softmax** operator. Note that other smoothing operators (*e.g.*, Mellowmax [2]) may also be considered in the future.

**Step 2: error control in updating $\mathcal{L}$.** Given the sub-optimality of the Boltzmann policy, one needs to characterize the difference between the optimal policy and the non-optimal ones. In particular, one can define the action gap between the best action and the second best action in terms of the Q-value as $\delta^s(\mathcal{L}) := \max_{a' \in \mathcal{A}} Q_{\mathcal{L}}^{\star}(s, a') - \max_{a \notin \text{argmax}_{a \in \mathcal{A}} Q_{\mathcal{L}}^{\star}(s,a)} Q_{\mathcal{L}}^{\star}(s,a) > 0$. Action gap is important for approximation algorithms [3], and are closely related to the problem-dependent bounds for regret analysis in reinforcement learning and multi-armed bandits, and advantage learning algorithms including A2C [27].

The problem is: in order for the learning algorithm to converge in terms of $\mathcal{L}$ (Theorem 2), one needs to ensure a definite differentiation between the optimal policy and the sub-optimal ones. This is problematic as the infimum of $\delta^s(\mathcal{L})$ over an infinite number of $\mathcal{L}$ can be 0. To address this, the population distribution at step $k$, say $\mathcal{L}_k$, needs to be projected to a finite grid, called $\epsilon$-net. The relation between the $\epsilon$-net and action gaps is as follows:

*For any $\epsilon > 0$, there exist a positive function $\phi(\epsilon)$ and an $\epsilon$-net $S_\epsilon := \{\mathcal{L}^{(1)}, \dots, \mathcal{L}^{(N_\epsilon)}\} \subseteq \mathcal{P}(\mathcal{S} \times \mathcal{A})$, with the properties that $\min_{i=1,\dots,N_\epsilon} d_{TV}(\mathcal{L}, \mathcal{L}^{(i)}) \leq \epsilon$ for any $\mathcal{L} \in \mathcal{P}(\mathcal{S} \times \mathcal{A})$, and that $\max_{a' \in \mathcal{A}} Q_{\mathcal{L}^{(i)}}^{\star}(s, a') - Q_{\mathcal{L}^{(i)}}^{\star}(s, a) \geq \phi(\epsilon)$ for any $i = 1, \dots, N_\epsilon$, $s \in \mathcal{S}$, and any $a \notin argmax_{a \in \mathcal{A}} Q_{\mathcal{L}^{(i)}}^{\star}(s, a)$.*

Here the existence of $\epsilon$-nets is trivial due to the compactness of the probability simplex $\mathcal{P}(\mathcal{S} \times \mathcal{A})$, and the existence of $\phi(\epsilon)$ comes from the finiteness of the action set $\mathcal{A}$. In practice, $\phi(\epsilon)$ often takes the form of $D\epsilon^\alpha$ with $D > 0$ and the exponent $\alpha > 0$ characterizing the decay rate of the action gaps.

Finally, to enable Q-learning, it is assumed that one has access to a population simulator (See [30, 38]). That is, for any policy $\pi \in \Pi$, given the current state $s \in \mathcal{S}$, for any population distribution $\mathcal{L}$, one can obtain the next state $s' \sim P(\cdot | s, \pi(s, \mu), \mathcal{L})$, a reward $r = r(s, \pi(s, \mu), \mathcal{L})$, and the next population distribution $\mathcal{L}' = \mathbb{P}_{s', \pi(s', \mu)}$. For brevity, we denote the simulator as $(s', r, \mathcal{L}') = \mathcal{G}(s, \pi, \mathcal{L})$. Here $\mu$ is the state marginal distribution of $\mathcal{L}$.

In summary, we propose the following Algorithm 1.

---
**Algorithm 1 Q-learning for GMFGs (GMF-Q)**

---
1: **Input**: Initial $\mathcal{L}_0$, tolerance $\epsilon > 0$.
2: **for** $k = 0, 1, \cdots$ **do**
3:    Perform Q-learning for $T_k$ iterations to find the approximate Q-function $\hat{Q}_k^{\star}(s, a) = \hat{Q}_{\mathcal{L}_k}^{\star}(s, a)$ of an MDP with dynamics $P_{\mathcal{L}_k}(s'|s, a)$ and rewards $r_{\mathcal{L}_k}(s, a)$.
4:    Compute $\pi_k \in \Pi$ with $\pi_k(s) = \textbf{softmax}_c(\hat{Q}_k^{\star}(s, \cdot))$.
5:    Sample $s \sim \mu_k$ ($\mu_k$ is the population state marginal of $\mathcal{L}_k$), obtain $\tilde{\mathcal{L}}_{k+1}$ from $\mathcal{G}(s, \pi_k, \mathcal{L}_k)$.
6:    Find $\mathcal{L}_{k+1} = \textbf{Proj}_{S_\epsilon}(\tilde{\mathcal{L}}_{k+1})$
7: **end for**

---

Note that **softmax** is applied only at the end of each outer iteration when a good approximation of $Q$ function is obtained. Within the outer iteration for the MDP problem with fixed mean-field information, standard Q-learning method is applied.

Here $\mathbf{Proj}_{S_\epsilon}(\mathcal{L}) = \arg\min_{\mathcal{L}^{(1)},\dots,\mathcal{L}^{(N_\epsilon)}} d_{TV}(\mathcal{L}^{(i)}, \mathcal{L})$. For computational tractability, it would be sufficient to choose $S_\epsilon$ as a truncation grid so that projection of $\tilde{\mathcal{L}}_k$ onto the epsilon-net reduces to truncating $\tilde{\mathcal{L}}_k$ to a certain number of digits. For instance, in our experiment, the number of digits is chosen to be 4. The choices of the hyper-parameters $c$ and $T_k$ can be found in Lemma 8 and Theorem 2. In practice, the algorithm is rather robust with respect to these hyper-parameters.

In the special case when the rewards $r_{\mathcal{L}}$ and transition dynamics $P(\cdot|s, a, \mathcal{L})$ are known, one can replace the Q-learning step in the above Algorithm 1 by a value iteration, resulting in the GMF-V Algorithm 3 in the Appendix.

We next show the convergence of this GMF-Q algorithm (Algorithm 1) to an $\epsilon$-Nash of (GMFG), with complexity analysis.

**Theorem 2** (Convergence and complexity of GMF-Q)**.** *Assume the same conditions in Theorem 4 and Lemma 8 in the Appendix. For any tolerances $\epsilon$, $\delta > 0$, set $\delta_k = \delta/K_{\epsilon,\eta}$, $\epsilon_k = (k+1)^{-(1+\eta)}$ for some $\eta \in (0, 1]$ $(k = 0, \dots, K_{\epsilon,\eta} - 1)$, $T_k = T^{\mathcal{M}_{\mathcal{L}_k}}(\delta_k, \epsilon_k)$ (defined in Lemma 8 in the Appendix) and $c = \frac{\log(1/\epsilon)}{\phi(\epsilon)}$. Then with probability at least $1 - 2\delta$, $W_1(\mathcal{L}_{K_{\epsilon,\eta}}, \mathcal{L}^\star) \leq C\epsilon$.*

*Moreover, the total number of iterations $T = \sum_{k=0}^{K_{\epsilon,\eta}-1} T^{\mathcal{M}_{\mathcal{L}_k}}(\delta_k, \epsilon_k)$ is bounded by [3]*

$$T = O\left(K_{\epsilon,\eta}^{1+\frac{4}{h}} \left(\log(K_{\epsilon,\eta}/\delta)\right)^{\frac{2}{1-h}+\frac{2}{h}+3}\right). \tag{11}$$

*Here $K_{\epsilon,\eta} := \left\lceil 2\max\left\{(\eta\epsilon/c)^{-1/\eta}, \log_d(\epsilon/\max\{diam(\mathcal{S})diam(\mathcal{A}), c\}) + 1\right\}\right\rceil$ is the number of outer iterations, $h$ is the step-size exponent in Q-learning (defined in Lemma 8 in the Appendix), and the constant $C$ is independent of $\delta$, $\epsilon$ and $\eta$.*

The proof of Theorem 2 in the Appendix depends on the Lipschitz continuity of the **softmax** operator [8], the closeness between **softmax** and the **argmax-e** (Lemma 7 in the Appendix), and the complexity of Q-learning for the MDP (Lemma 8 in the Appendix).

## 5 Experiment: repeated auction game

In this section, we report the performance of the proposed GMF-Q Algorithm. The objectives of the experiments include 1) testing the convergence, stability, and learning ability of GMF-Q in the GMFG setting, and 2) comparing GMF-Q with existing multi-agent reinforcement learning algorithms, including IL algorithm and MF-Q algorithm.

We take the GMFG framework for the repeated auction game from Section 2.3. Here each advertiser learns to bid in the auction with a budget constraint.

**Parameters.** The model parameters are set as: $|\mathcal{S}| = |\mathcal{A}| = 10$, the overbidding penalty $\rho = 0.2$, the distributions of the conversion rate $v \sim \text{uniform}(\{1, 2, 3, 4\})$, and the competition intensity index $M = 5$. The random fulfillment is chosen as: if $s < s_{\max}$, $\Delta(s) = 1$ with probability $\frac{1}{2}$ and $\Delta(s) = 0$ with probability $\frac{1}{2}$; if $s = s_{\max}$, $\Delta(s) = 0$.

The algorithm parameters are (unless otherwise specified): the temperature parameter $c = 4.0$, the discount factor $\gamma = 0.8$, the parameter $h$ from Lemma 8 in the Appendix being $h = 0.87$, and the baseline inner iteration being 2000. Recall that for GMF-Q, both $v$ and the dynamics of $P$ for $s$ are unknown *a priori*. The 90%-confidence intervals are calculated with 20 sample paths.

**Performance evaluation in the GMFG setting.** Our experiment shows that the GMF-Q Algorithm is efficient and robust, and learns well.

*Convergence and stability of GMF-Q.* GMF-Q is efficient and robust. First, GMF-Q converges after about 10 outer iterations; secondly, as the number of inner iterations increases, the error decreases (Figure 2); and finally, the convergence is robust with respect to both the change of number of states and the initial population distribution (Figure 3).

In contrast, the Naive algorithm does not converge even with 10000 inner iterations, and the joint distribution $\mathcal{L}_t$ keeps fluctuating (Figure 4).

Table 1: Q-table with $T_k^{\text{GMF-V}} = 5000$.

| $T_k^{\text{GMF-Q}}$ | 1000 | 3000 | 5000 | 10000 |
|---|---|---|---|---|
| $\Delta Q$ | 0.21263 | 0.1294 | 0.10258 | 0.0989 |

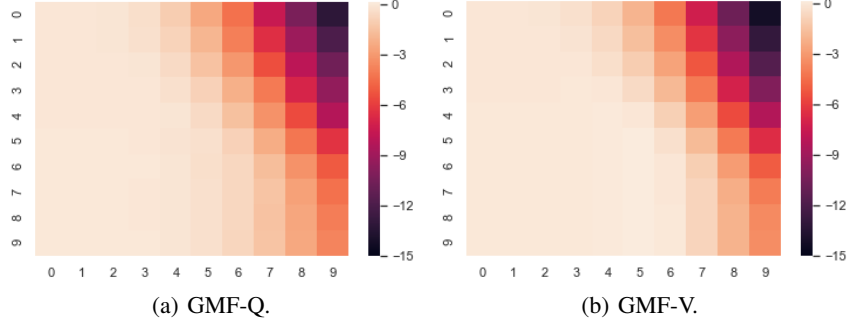

(a) GMF-Q.　　　　　　　　　　(b) GMF-V.

Figure 1: Q-tables: GMF-Q vs. GMF-V.

*Learning accuracy of GMF-Q.* GMF-Q learns well. Its learning accuracy is tested against its special form GMF-V (Appendix G), with the latter assuming a known distribution of conversion rate $v$ and the dynamics $P$ for the budget $s$. The relative $L_2$ distance between the Q-tables of these two algorithms is $\Delta Q := \frac{\|Q_{\text{GMF-V}} - Q_{\text{GMF-Q}}\|_2}{\|Q_{\text{GMF-V}}\|_2} = 0.098879$. This implies that GMF-Q learns the true GMFG solution with 90-percent accuracy with 10000 inner iterations.

The heatmap in Figure 1(a) is the Q-table for GMF-Q Algorithm after 20 outer iterations. Within each outer iteration, there are $T_k^{\text{GMF-Q}} = 10000$ inner iterations. The heatmap in Figure 1(b) is the Q-table for GMF-Q Algorithm after 20 outer iterations. Within each outer iteration, there are $T_k^{\text{GMF-V}} = 5000$ inner iterations.

**Comparison with existing algorithms for $N$-player games.** To test the effectiveness of GMF-Q for approximating $N$-player games, we next compare GMF-Q with IL algorithm and MF-Q algorithm. IL algorithm [36] considers $N$ independent players and each player solves a decentralized reinforcement learning problem ignoring other players in the system. The MF-Q algorithm [40] extends the NASH-Q Learning algorithm for the $N$-player game introduced in [15], adds the aggregate actions ($\bar{\boldsymbol{a}}_{-i} = \frac{\sum_{j \neq i} a_j}{N-1}$) from the opponents, and works for the class of games where the interactions are only through the average actions of $N$ players.

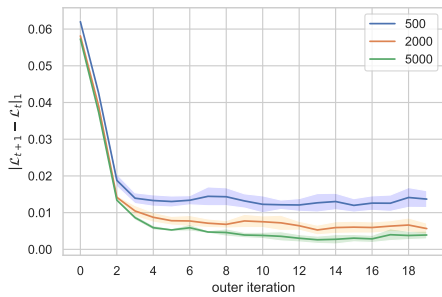

Figure 2: Convergence with different number of inner iterations.

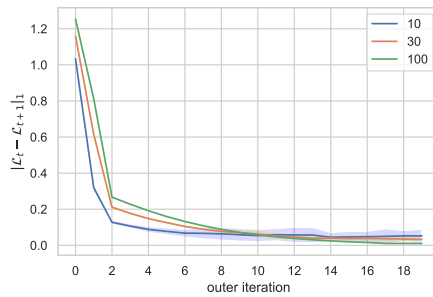

Figure 3: Convergence with different number of states.

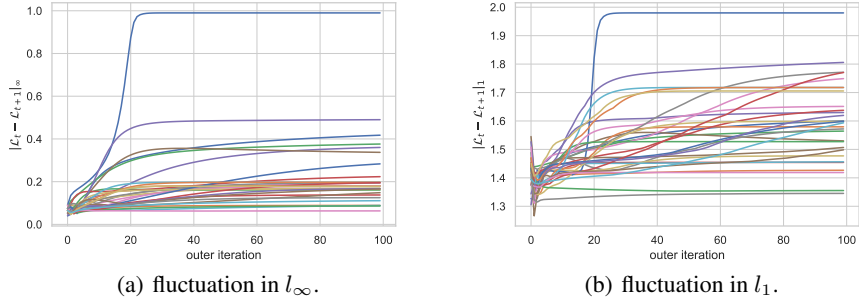

(a) fluctuation in $l_\infty$.    (b) fluctuation in $l_1$.

Figure 4: Fluctuations of Naive Algorithm (30 sample paths).

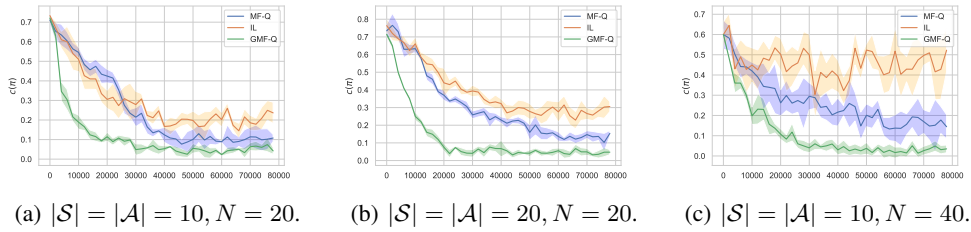

(a) $|\mathcal{S}| = |\mathcal{A}| = 10, N = 20$.    (b) $|\mathcal{S}| = |\mathcal{A}| = 20, N = 20$.    (c) $|\mathcal{S}| = |\mathcal{A}| = 10, N = 40$.

Figure 5: Learning accuracy based on $C(\boldsymbol{\pi})$.

*Performance metric.* We adopt the following metric to measure the difference between a given policy $\pi$ and an NE (here $\epsilon_0 > 0$ is a safeguard, and is taken as $0.1$ in the experiments):

$$C(\boldsymbol{\pi}) = \frac{1}{N|\mathcal{S}|^N} \sum_{i=1}^{N} \sum_{\boldsymbol{s} \in \mathcal{S}^N} \frac{\max_{\pi^i} V_i(\boldsymbol{s}, (\boldsymbol{\pi}^{-i}, \pi^i)) - V_i(\boldsymbol{s}, \boldsymbol{\pi})}{|\max_{\pi^i} V_i(\boldsymbol{s}, (\boldsymbol{\pi}^{-i}, \pi^i))| + \epsilon_0}.$$

Clearly $C(\boldsymbol{\pi}) \geq 0$, and $C(\boldsymbol{\pi}^*) = 0$ if and only if $\boldsymbol{\pi}^*$ is an NE. Policy $\arg\max_{\pi_i} V_i(\boldsymbol{s}, (\boldsymbol{\pi}^{-i}, \pi_i))$ is called the best response to $\boldsymbol{\pi}^{-i}$. A similar metric without normalization has been adopted in [29].

Our experiment (Figure 5) shows that GMF-Q is superior in terms of convergence rate, accuracy, and stability for approximating an $N$-player game: GMF-Q converges faster than IL and MF-Q, with the smallest error, and with the lowest variance, as $\epsilon$-net improves the stability.

For instance, when $N = 20$, IL Algorithm converges with the largest error $0.220$. The error from MF-Q is $0.101$, smaller than IL but still bigger than the error from GMF-Q. The GMF-Q converges with the lowest error $0.065$. Moreover, as $N$ increases, the error of GMF-Q deceases while the errors of both MF-Q and IL increase significantly. As $|\mathcal{S}|$ and $|\mathcal{A}|$ increase, GMF-Q is robust with respect to this increase of dimensionality, while both MF-Q and IL clearly suffer from the increase of the dimensionality with decreased convergence rate and accuracy. Therefore, GMF-Q is more scalable than IL and MF-Q, when the system is complex and the number of players $N$ is large.

## 6 Conclusion

This paper builds a GMFG framework for simultaneous learning and decision-making, establishes the existence and uniqueness of NE, and proposes a Q-learning algorithm GMF-Q with convergence and complexity analysis. Experiments demonstrate superior performance of GMF-Q.

## Acknowledgment

We thank Haoran Tang for the insightful early discussion on stabilizing the Q-learning algorithm and sharing the ideas of his work on soft-Q-learning [12], which motivates our adoption of the soft-max operators. We also thank the anonymous NeurIPS 2019 reviewers for the valuable suggestions.

## Footnotes

[1] Here the indicator function $\mathbf{I}_{s_t^j = s} = 1$ if $s_t^j = s$ and 0 otherwise.

[2]**argmax-e** is not continuous: Let $x = (1, 1)$, then **argmax-e**$(x) = (1/2, 1/2)$. For any $\epsilon > 0$, let $y = (1, 1 - \epsilon)$, then **argmax-e**$(y) = (1, 0)$.

[3]Let $h = \frac{3}{4}$, $\eta = 1$, the bound reduces to $T = O(K_\epsilon^{\frac{19}{3}} (\log(\frac{K_\epsilon}{\delta}))^{\frac{41}{3}})$. Note that this bound may not be tight.

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
