[Supplementary Material]

## A    Distance metrics and completeness

This section reviews some basic properties of the Wasserstein distance. It then proves that the metrics defined in the main text are indeed distance functions and define complete metric spaces.

$\ell_1$**-Wasserstein distance and dual representation.**    The $\ell_1$ Wasserstein distance over $\mathcal{P}(\mathcal{X})$ for $\mathcal{X} \subseteq \mathbb{R}^k$ is defined as

$$W_1(\nu, \nu') := \inf_{M \in \mathcal{M}(\nu, \nu')} \int_{\mathcal{X} \times \mathcal{X}} \|x - y\|_2 dM(x, y). \tag{12}$$

where $\mathcal{M}(\nu, \nu')$ is the set of all measures (couplings) on $\mathcal{X} \times \mathcal{X}$, with marginals $\nu$ and $\nu'$ on the two components, respectively.

The Kantorovich duality theorem enables the following equivalent dual representation of $W_1$:

$$W_1(\nu, \nu') = \sup_{\|f\|_L \leq 1} \left| \int_{\mathcal{X}} f d\nu - \int_{\mathcal{X}} f d\nu' \right|, \tag{13}$$

where the supremum is taken over all 1-Lipschitz functions $f$, *i.e.*, $f$ satisfying $|f(x) - f(y)| \leq \|x - y\|_2$ for all $x, y \in \mathcal{X}$.

The Wasserstein distance $W_1$ can also be related to the total variation distance via the following inequalities [9]:

$$d_{\min}(\mathcal{X})d_{TV}(\nu, \nu') \leq W_1(\nu, \nu') \leq \text{diam}(\mathcal{X})d_{TV}(\nu, \nu'), \tag{14}$$

where $d_{\min}(\mathcal{X}) = \min_{x \neq y \in \mathcal{X}} \|x - y\|_2$, which is guaranteed to be positive when $\mathcal{X}$ is finite.

When $\mathcal{S}$ and $\mathcal{A}$ are compact, for any compact subset $\mathcal{X} \subseteq \mathbb{R}^k$, and for any $\nu, \nu' \in \mathcal{P}(\mathcal{X})$, $W_1(\nu, \nu') \leq \text{diam}(\mathcal{X})d_{TV}(\nu, \nu') \leq \text{diam}(\mathcal{X}) < \infty$, where $\text{diam}(\mathcal{X}) = \sup_{x,y \in \mathcal{X}} \|x - y\|_2$ and $d_{TV}$ is the total variation distance. Moreover, one can verify

**Lemma 3.** *Both $D$ and $\mathcal{W}_1$ are distance functions, and they are finite for any input distribution pairs. In addition, both $(\{\Pi\}_{t=0}^\infty, D)$ and $(\{\mathcal{P}(\mathcal{S} \times \mathcal{A})\}_{t=0}^\infty, \mathcal{W}_1)$ are complete metric spaces.*

These facts enable the usage of Banach fixed-point mapping theorem for the proof of existence and uniqueness (Theorems 1 and 4).

*Proof of Lemma 3.*  It is known that for any compact set $\mathcal{X} \subseteq \mathbb{R}^k$, $(\mathcal{P}(\mathcal{X}), W_1)$ defines a complete metric space [5]. Since $W_1(\nu, \nu') \leq \text{diam}(\mathcal{X})$ is uniformly bounded for any $\nu, \nu' \in \mathcal{P}(\mathcal{X})$, we know that $\mathcal{W}_1(\mathcal{L}, \mathcal{L}') \leq \text{diam}(\mathcal{X})$ and $D(\boldsymbol{\pi}, \boldsymbol{\pi}') \leq \text{diam}(\mathcal{X})$ as well, so they are both finite for any input distribution pairs. It is clear that they are distance functions based on the fact that $W_1$ is a distance function.

Finally, we show the completeness of the two metric spaces $(\{\Pi\}_{t=0}^\infty, D)$ and $(\{\mathcal{P}(\mathcal{S} \times \mathcal{A})\}_{t=0}^\infty, \mathcal{W}_1)$. Take $(\{\Pi\}_{t=0}^\infty, D)$ for example. Suppose that $\boldsymbol{\pi}^k$ is a Cauchy sequence in $(\{\Pi\}_{t=0}^\infty, D)$. Then for any $\epsilon > 0$, there exists a positive integer $N$, such that for any $m, n \geq N$,

$$D(\boldsymbol{\pi}^n, \boldsymbol{\pi}^m) \leq \epsilon \implies W_1(\pi_t^n(s), \pi_t^m(s)) \leq \epsilon \text{ for any } s \in \mathcal{S}, t \in \mathbb{N}, \tag{15}$$

which implies that $\pi_t^k(s)$ forms a Cauchy sequence in $(\mathcal{P}(\mathcal{A}), W_1)$, and hence by the completeness of $(\mathcal{P}(\mathcal{A}), W_1)$, $\pi_t^k(s)$ converges to some $\pi_t(s) \in \mathcal{P}(\mathcal{A})$. As a result, $\boldsymbol{\pi}^n \to \boldsymbol{\pi} \in \{\Pi\}_{t=0}^\infty$ under metric $D$, which shows that $(\{\Pi\}_{t=0}^\infty, D)$ is complete.

The completeness of $(\{\mathcal{P}(\mathcal{S} \times \mathcal{A})\}_{t=0}^\infty, \mathcal{W}_1)$ can be proved similarly.                $\square$

The same argument for Lemma 3 shows that both $D$ and $W_1$ are distance functions and are finite for any input distribution pairs, with both $(\Pi, D)$ and $(\mathcal{P}(\mathcal{S} \times \mathcal{A}), W_1)$ again complete metric spaces.

## B    Existence and uniqueness for stationary NE of GMFGs

**Definition B.1** (Stationary NE for GMFGs). *In* (GMFG)*, a player-population profile $(\pi^\star, \mathcal{L}^\star)$ is called a stationary NE if*

1. *(Single player side) For any policy $\pi$ and any initial state $s \in \mathcal{S}$,*

$$V(s, \pi^\star, \mathcal{L}^\star) \geq V(s, \pi, \mathcal{L}^\star). \tag{16}$$

2. *(Population side) $\mathbb{P}_{s_t, a_t} = \mathcal{L}^\star$ for all $t \geq 0$, where $\{s_t, a_t\}_{t=0}^{\infty}$ is the dynamics under the policy $\pi^\star$ starting from $s_0 \sim \mu^\star$, with $a_t \sim \pi^\star(s_t, \mu^\star)$, $s_{t+1} \sim P(\cdot|s_t, a_t, \mathcal{L}^\star)$, and $\mu^\star$ being the population state marginal of $\mathcal{L}^\star$.*

The existence and uniqueness of the NE to (GMFG) in the stationary setting can be established by modifying appropriately the same fixed-point approach for the GMFG in the main text.

**Step 1.** Fix $\mathcal{L}$, the GMFG becomes the classical optimization problem. That is, solving (GMFG) is now reduced to finding a policy $\pi_{\mathcal{L}}^\star \in \Pi := \{\pi \,|\, \pi : \mathcal{S} \to \mathcal{P}(\mathcal{A})\}$ to maximize

$$V(s, \pi_{\mathcal{L}}, \mathcal{L}) := \quad \mathbb{E}\left[\sum_{t=0}^{\infty} \gamma^t r(s_t, a_t, \mathcal{L})|s_0 = s\right],$$
$$\text{subject to} \quad s_{t+1} \sim P(s_t, a_t, \mathcal{L}), \quad a_t \sim \pi_{\mathcal{L}}(s_t).$$

Now given this fixed $\mathcal{L}$ and the solution $\pi_{\mathcal{L}}^\star$ to the above optimization problem, one can again define

$$\Gamma_1 : \mathcal{P}(\mathcal{S} \times \mathcal{A}) \to \Pi,$$

such that $\pi_{\mathcal{L}}^\star = \Gamma_1(\mathcal{L})$. Note that this $\pi_{\mathcal{L}}^\star$ satisfies the single player side condition for the population state-action pair $L$,

$$V(s, \pi_{\mathcal{L}}^\star, \mathcal{L}) \geq V(s, \pi, \mathcal{L}), \tag{17}$$

for any policy $\pi$ and any initial state $s \in \mathcal{S}$.

Accordingly, a similar feedback regularity condition is needed in this step.

**Assumption 3.** *There exists a constant $d_1 \geq 0$, such that for any $\mathcal{L}, \mathcal{L}' \in \mathcal{P}(\mathcal{S} \times \mathcal{A})$,*

$$D(\Gamma_1(\mathcal{L}), \Gamma_1(\mathcal{L}')) \leq d_1 W_1(\mathcal{L}, \mathcal{L}'), \tag{18}$$

*where*

$$D(\pi, \pi') := \sup_{s \in \mathcal{S}} W_1(\pi(s), \pi'(s)), \tag{19}$$

*and $W_1$ is the $\ell_1$-Wasserstein distance (a.k.a. earth mover distance) between probability measures.*

**Step 2.** Based on the analysis of Step 1 and $\pi_{\mathcal{L}}^\star$, update the initial $\mathcal{L}$ to $\mathcal{L}'$ following the controlled dynamics $P(\cdot|s_t, a_t, \mathcal{L})$.

Accordingly, define a mapping $\Gamma_2 : \Pi \times \mathcal{P}(\mathcal{S} \times \mathcal{A}) \to \mathcal{P}(\mathcal{S} \times \mathcal{A})$ as follows:

$$\Gamma_2(\pi, \mathcal{L}) := \hat{\mathcal{L}} = \mathbb{P}_{s_1, a_1}, \tag{20}$$

where $a_1 \sim \pi(s_1)$, $s_1 \sim \mu P(\cdot|\cdot, a_0, \mathcal{L})$, $a_0 \sim \pi(s_0)$, $s_0 \sim \mu$, and $\mu$ is the population state marginal of $\mathcal{L}$.

One also needs a similar assumption in this step.

**Assumption 4.** *There exist constants $d_2$, $d_3 \geq 0$, such that for any admissible policies $\pi, \pi_1, \pi_2$ and joint distributions $\mathcal{L}, \mathcal{L}_1, \mathcal{L}_2$,*

$$W_1(\Gamma_2(\pi_1, \mathcal{L}), \Gamma_2(\pi_2, \mathcal{L})) \leq d_2 D(\pi_1, \pi_2), \tag{21}$$

$$W_1(\Gamma_2(\pi, \mathcal{L}_1), \Gamma_2(\pi, \mathcal{L}_2)) \leq d_3 W_1(\mathcal{L}_1, \mathcal{L}_2). \tag{22}$$

**Step 3.** Repeat until $\mathcal{L}'$ matches $\mathcal{L}$.

This step is to ensure the population side condition. To ensure the convergence of the combined step one and step two, it suffices if $\Gamma : \mathcal{P}(\mathcal{S} \times \mathcal{A}) \to \mathcal{P}(\mathcal{S} \times \mathcal{A})$ with $\Gamma(\mathcal{L}) := \Gamma_2(\Gamma_1(\mathcal{L}), \mathcal{L})$ is a contractive mapping (under the $W_1$ distance).

Similar to the proof of Theorem 1, again by the Banach fixed point theorem and the completeness of the related metric spaces, there exists a unique stationary NE of the GMFG. That is,

**Theorem 4** (Existence and Uniqueness of stationary MFG solution). *Given Assumptions 3 and 4, and assume $d_1 d_2 + d_3 < 1$. Then there exists a unique stationary NE to* (GMFG).

## C  Additional comments on assumptions

As mentioned in the main text, the single player side Assumption 1 and its counterpart Assumption 3 for the stationary version correspond to the feedback regularity condition in the classical MFG literature. Here we add some comments on the population side Assumption 2 and its stationary version Assumption 4. For simplicity and clarity, let us consider the stationary case with finite state and action spaces. Then we have the following result.

**Lemma 5.** *Suppose that* $\max_{s,a,\mathcal{L},s'} P(s'|s,a,\mathcal{L}) \leq c_1$, *and that* $P(s'|s,a,\cdot)$ *is* $c_2$-*Lipschitz in* $W_1$, *i.e.,*

$$|P(s'|s,a,\mathcal{L}_1) - P(s'|s,a,\mathcal{L}_2)| \leq c_2 W_1(\mathcal{L}_1, \mathcal{L}_2). \tag{23}$$

*Then in Assumption 4,* $d_2$ *and* $d_3$ *can be chosen as*

$$d_2 = \frac{2diam(\mathcal{S})diam(\mathcal{A})|\mathcal{S}|c_1}{d_{\min}(\mathcal{A})} \tag{24}$$

*and* $d_3 = \frac{diam(\mathcal{S})diam(\mathcal{A})c_2}{2}$, *respectively.*

Lemma 5 provides an explicit characterization of the population side assumptions based only on the boundedness and Lipschitz properties of the transition dynamics $P$. In particular, $c_1$ becomes smaller when the transition dynamics becomes more diverse and the state space becomes larger.

*Proof.* (Lemma 5) We begin by noticing that $\mathcal{L}' = \Gamma_2(\pi, \mathcal{L})$ can be expanded and computed as follows:

$$\mu'(s') = \sum_{s\in\mathcal{S},a\in\mathcal{A}} \mu(s)P(s'|s,a,\mathcal{L})\pi(s,a), \quad \mathcal{L}'(s',a') = \mu'(s')\pi(s',a'), \tag{25}$$

where $\mu$ is the state marginal distribution of $\mathcal{L}$.

Now by the inequalities (14), we have

$$W_1(\Gamma_2(\pi_1,\mathcal{L}), \Gamma_2(\pi_2,\mathcal{L})) \leq diam(\mathcal{S}\times\mathcal{A})d_{TV}(\Gamma_2(\pi_1,\mathcal{L}), \Gamma_2(\pi_2,\mathcal{L}))$$

$$=\frac{diam(\mathcal{S}\times\mathcal{A})}{2}\sum_{s'\in\mathcal{S},a'\in\mathcal{A}}\left|\sum_{s\in\mathcal{S},a\in\mathcal{A}}\mu(s)P(s'|s,a,\mathcal{L})\left(\pi_1(s,a)\pi_1(s',a') - \pi_2(s,a)\pi_2(s',a')\right)\right|$$

$$\leq\frac{diam(\mathcal{S}\times\mathcal{A})}{2}\max_{s,a,\mathcal{L},s'}P(s'|s,a,\mathcal{L})\sum_{s,a,s',a'}\mu(s)(\pi_1(s,a) + \pi_2(s,a))|\pi_1(s',a') - \pi_2(s',a')|$$

$$\leq\frac{diam(\mathcal{S}\times\mathcal{A})}{2}\max_{s,a,\mathcal{L},s'}P(s'|s,a,\mathcal{L})\sum_{s',a'}|\pi_1(s',a') - \pi_2(s',a')|\cdot(1+1)$$

$$=2diam(\mathcal{S}\times\mathcal{A})\max_{s,a,\mathcal{L},s'}P(s'|s,a,\mathcal{L})\sum_{s'}d_{TV}(\pi_1(s'), \pi_2(s'))$$

$$\leq\frac{2diam(\mathcal{S}\times\mathcal{A})\max_{s,a,\mathcal{L},s'}P(s'|s,a,\mathcal{L})|\mathcal{S}|}{d_{\min}(\mathcal{A})}D(\pi_1,\pi_2) = \frac{2diam(\mathcal{S})diam(\mathcal{A})|\mathcal{S}|c_1}{d_{\min}(\mathcal{A})}D(\pi_1,\pi_2). \tag{26}$$

Similarly, we have

$$W_1(\Gamma_2(\pi,\mathcal{L}_1), \Gamma_2(\pi,\mathcal{L}_2)) \leq diam(\mathcal{S}\times\mathcal{A})d_{TV}(\Gamma_2(\pi,\mathcal{L}_1), \Gamma_2(\pi,\mathcal{L}_2))$$

$$=\frac{diam(\mathcal{S}\times\mathcal{A})}{2}\sum_{s'\in\mathcal{S},a'\in\mathcal{A}}\left|\sum_{s\in\mathcal{S},a\in\mathcal{A}}\mu(s)\pi(s,a)\pi(s',a')\left(P(s'|s,a,\mathcal{L}_1) - P(s'|s,a,\mathcal{L}_2)\right)\right|$$

$$\leq\frac{diam(\mathcal{S}\times\mathcal{A})}{2}\sum_{s,a,s',a'}\mu(s)\pi(s,a)\pi(s',a')\left|P(s'|s,a,\mathcal{L}_1) - P(s'|s,a,\mathcal{L}_2)\right| \tag{27}$$

$$\leq\frac{diam(\mathcal{S})diam(\mathcal{A})c_2}{2}.$$

This completes the proof. □

# D  Proof of Theorems 1 and 4

For notational simplicity, we only present the proof for the stationary case (Theorem 4). The proof of Theorems 1 is the same with appropriate notational changes.

First by Definition B.1 and the definitions of $\Gamma_i$ $(i = 1, 2)$, $(\pi, \mathcal{L})$ is a stationary NE iff $\mathcal{L} = \Gamma(\mathcal{L}) = \Gamma_2(\Gamma_1(\mathcal{L}), \mathcal{L})$ and $\pi = \Gamma_1(\mathcal{L})$, where $\Gamma(\mathcal{L}) = \Gamma_2(\Gamma_1(\mathcal{L}), \mathcal{L})$. This indicates that for any $\mathcal{L}_1, \mathcal{L}_2 \in \mathcal{P}(\mathcal{S} \times \mathcal{A})$,

$$W_1(\Gamma(\mathcal{L}_1), \Gamma(\mathcal{L}_2)) = W_1(\Gamma_2(\Gamma_1(\mathcal{L}_1), \mathcal{L}_1), \Gamma_2(\Gamma_1(\mathcal{L}_2), \mathcal{L}_2))$$
$$\leq W_1(\Gamma_2(\Gamma_1(\mathcal{L}_1), \mathcal{L}_1), \Gamma_2(\Gamma_1(\mathcal{L}_2), \mathcal{L}_1)) + W_1(\Gamma_2(\Gamma_1(\mathcal{L}_2), \mathcal{L}_1), \Gamma_2(\Gamma_1(\mathcal{L}_2), \mathcal{L}_2)) \quad (28)$$
$$\leq (d_1 d_2 + d_3) W_1(\mathcal{L}_1, \mathcal{L}_2).$$

And since $d_1 d_2 + d_3 \in [0, 1)$, by the Banach fixed-point theorem, we conclude that there exists a unique fixed-point of $\Gamma$, or equivalently, a unique stationary MFG solution to (GMFG).

# E  Proof of Theorem 2

The proof of Theorem 2 relies on the following lemmas.

**Lemma 6** ([8]). *The softmax function is $c$-Lipschitz, i.e., $\|\textbf{softmax}_c(x) - \textbf{softmax}_c(y)\|_2 \leq c\|x - y\|_2$ for any $x, y \in \mathbb{R}^n$.*

Notice that for a finite set $\mathcal{X} \subseteq \mathbb{R}^k$ and any two (discrete) distributions $\nu, \nu'$ over $\mathcal{X}$, we have

$$W_1(\nu, \nu') \leq \text{diam}(\mathcal{X}) d_{TV}(\nu, \nu') = \frac{\text{diam}(\mathcal{X})}{2} \|\nu - \nu'\|_1 \leq \frac{\text{diam}(\mathcal{X})}{2} \|\nu - \nu'\|_2, \quad (29)$$

where in computing the $\ell_1$-norm, $\nu, \nu'$ are viewed as vectors of length $|\mathcal{X}|$.

Hence Lemma 6 implies that for any $x, y \in \mathbb{R}^{|\mathcal{X}|}$, when $\textbf{softmax}_c(x)$ and $\textbf{softmax}_c(y)$ are viewed as probability distributions over $\mathcal{X}$, we have

$$W_1(\textbf{softmax}_c(x), \textbf{softmax}_c(y)) \leq \frac{\text{diam}(\mathcal{X})c}{2} \|x - y\|_2 \leq \frac{\text{diam}(\mathcal{X})\sqrt{|\mathcal{X}|}c}{2} \|x - y\|_\infty.$$

**Lemma 7.** *The distance between the softmax and the argmax mapping is bounded by*
$$\|\textbf{softmax}_c(x) - \textbf{argmax-e}(x)\|_2 \leq 2n \exp(-c\delta),$$
*where $\delta = x_{\max} - \max_{x_j < x_{\max}} x_j$, $x_{\max} = \max_{i=1,\ldots,n} x_i$, and $\delta := \infty$ when all $x_j$ are equal.*

Similar to Lemma 6, Lemma 7 implies that for any $x \in \mathbb{R}^{|\mathcal{X}|}$, viewing $\textbf{softmax}_c(x)$ as probability distributions over $\mathcal{X}$ leads to

$$W_1(\textbf{softmax}_c(x), \textbf{argmax-e}(x)) \leq \text{diam}(\mathcal{X})|\mathcal{X}| \exp(-c\delta).$$

*Proof of Lemma 7.* Without loss of generality, assume that $x_1 = x_2 = \cdots = x_m = \max_{i=1,\ldots,n} x_i = x^\star > x_j$ for all $m < j \leq n$. Then

$$\textbf{argmax-e}(x)_i = \begin{cases} \frac{1}{m}, & i \leq m, \\ 0, & otherwise. \end{cases}$$

$$\textbf{softmax}_c(x)_i = \begin{cases} \frac{e^{cx^\star}}{me^{cx^\star} + \sum_{j=m+1}^n e^{cx_j}}, & i \leq m, \\ \frac{e^{cx_i}}{me^{cx^\star} + \sum_{j=m+1}^n e^{cx_j}}, & otherwise. \end{cases}$$

Therefore

$$\|\textbf{softmax}_c(x) - \textbf{argmax-e}(x)\|_2 \leq \|\textbf{softmax}_c(x) - \textbf{argmax-e}(x)\|_1$$

$$= m\left(\frac{1}{m} - \frac{e^{cx^\star}}{me^{cx^\star} + \sum_{j=m+1}^n e^{cx_j}}\right) + \frac{\sum_{i=m+1}^n e^{cx_i}}{me^{cx^\star} + \sum_{j=m+1}^n e^{cx_j}}$$

$$= \frac{2\sum_{i=m+1}^n e^{cx_i}}{me^{cx^\star} + \sum_{i=m+1}^n e^{cx_i}} = \frac{2\sum_{i=m+1}^n e^{-c\delta_i}}{m + \sum_{i=m+1}^n e^{-c\delta_i}}$$

$$\leq \frac{2}{m}\sum_{i=m+1}^n e^{-c\delta_i} \leq \frac{2(n - m)}{m} e^{-c\delta} \leq 2n e^{-c\delta},$$

with $\delta_i = x_i - x^\star$. $\qquad\qquad\qquad\qquad\qquad\qquad\qquad\qquad\qquad\qquad\qquad\qquad\qquad$ $\square$

**Lemma 8** ([7]). *For an MDP, say $\mathcal{M}$, suppose that the Q-learning algorithm takes step-sizes*

$$\beta_t(s,a) = \begin{cases} |\#(s,a,t)+1|^{-h}, & (s,a) = (s_t, a_t), \\ 0, & otherwise. \end{cases}$$

*with $h \in (1/2, 1)$. Here $\#(s,a,t)$ is the number of times up to time $t$ that one visits the state-action pair $(s,a)$. Also suppose that the covering time of the state-action pairs is bounded by $L$ with probability at least $1 - p$ for some $p \in (0,1)$. Then $\|Q_{T^{\mathcal{M}}(\delta,\epsilon)} - Q^\star\|_\infty \leq \epsilon$ with probability at least $1 - 2\delta$. Here $Q_T$ is the $T$-th update in Q-learning, and $Q^\star$ is the (optimal) Q-function, given that*

$$T^{\mathcal{M}}(\delta,\epsilon) = \Omega\left( \left( \frac{L\log_p(\delta)}{\beta}\log\frac{V_{\max}}{\epsilon}\right)^{\frac{1}{1-h}} + \left( \frac{(L\log_p(\delta))^{1+3h}V_{\max}^2 \log\left(\frac{|\mathcal{S}||\mathcal{A}|V_{\max}}{\delta\beta\epsilon}\right)}{\beta^2\epsilon^2}\right)^{\frac{1}{h}}\right),$$

*where $\beta = (1-\gamma)/2$, $V_{\max} = R_{\max}/(1-\gamma)$, and $R_{\max}$ is an upper bound on the extreme difference between the expected rewards, i.e., $\max_{s,a,\mu} r(s,a,\mu) - \min_{s,a,\mu} r(s,a,\mu) \leq R_{\max}$.*

Here the covering time $L$ of a state-action pair sequence is defined to be the number of steps needed to visit all state-action pairs starting from any arbitrary state-action pair, and $T^{\mathcal{M}}(\delta,\epsilon)$ is the number of inner iterations $T_k$ set in Algorithm 1. This will guarantee the convergence in Theorem 2. Also notice that the $l_\infty$ norm above is defined in an element-wise sense, *i.e.*, for $M \in \mathbb{R}^{|\mathcal{S}|\times|\mathcal{A}|}$, we have $\|M\|_\infty = \max_{s\in\mathcal{S}, a\in\mathcal{A}} |M(s,a)|$.

*Proof of Theorem 2.* Define $\hat{\Gamma}_1^k(\mathcal{L}_k) := \mathbf{softmax}_c\left(\hat{Q}_{\mathcal{L}_k}^\star\right)$. In the following, $\pi = \mathbf{softmax}_c(Q_\mathcal{L})$ is understood as the policy $\pi$ with $\pi(s) = \mathbf{softmax}_c(Q_\mathcal{L}(s,\cdot))$. Let $\mathcal{L}^\star$ be the population state-action pair in a stationary NE of (GMFG). Then $\pi_k = \hat{\Gamma}_1^k(\mathcal{L}_k)$. Denoting $d := d_1 d_2 + d_3$, we see

$$\begin{aligned}
W_1(\tilde{\mathcal{L}}_{k+1}, \mathcal{L}^\star) &= W_1(\Gamma_2(\pi_k, \mathcal{L}_k), \Gamma_2(\Gamma_1(\mathcal{L}^\star), \mathcal{L}^\star)) \\
&\leq W_1(\Gamma_2(\Gamma_1(\mathcal{L}_k), \mathcal{L}_k), \Gamma_2(\Gamma_1(\mathcal{L}^\star), \mathcal{L}^\star)) + W_1(\Gamma_2(\Gamma_1(\mathcal{L}_k), \mathcal{L}_k), \Gamma_2(\hat{\Gamma}_1^k(\mathcal{L}_k), \mathcal{L}_k)) \\
&\leq W_1(\Gamma(\mathcal{L}_k), \Gamma(\mathcal{L}^\star)) + d_2 D(\Gamma_1(\mathcal{L}_k), \hat{\Gamma}_1^k(\mathcal{L}_k)) \\
&\leq (d_1 d_2 + d_3) W_1(\mathcal{L}_k, \mathcal{L}^\star) + d_2 D(\mathbf{argmax\text{-}e}(Q_{\mathcal{L}_k}^\star), \mathbf{softmax}_c(\hat{Q}_{\mathcal{L}_k}^\star)) \\
&\leq d W_1(\mathcal{L}_k, \mathcal{L}^\star) + d_2 D(\mathbf{softmax}_c(\hat{Q}_{\mathcal{L}_k}^\star), \mathbf{softmax}_c(Q_{\mathcal{L}_k}^\star)) \\
&\qquad + d_2 D(\mathbf{argmax\text{-}e}(Q_{\mathcal{L}_k}^\star), \mathbf{softmax}_c(Q_{\mathcal{L}_k}^\star)) \\
&\leq d W_1(\mathcal{L}_k, \mathcal{L}^\star) + \frac{cd_2\mathrm{diam}(\mathcal{A})\sqrt{|\mathcal{A}|}}{2}\|\hat{Q}_{\mu_k}^\star - Q_{\mu_k}^\star\|_\infty \\
&\qquad + d_2 D(\mathbf{argmax\text{-}e}(Q_{\mathcal{L}_k}^\star), \mathbf{softmax}_c(Q_{\mathcal{L}_k}^\star)).
\end{aligned}$$

Then since $\mathcal{L}_k \in S_\epsilon$ by the projection step, Lemma 7, and Lemma 8 with the choice of $T_k = T^{\mathcal{M}_\mu}(\delta_k, \epsilon_k))$, we have, with probability at least $1 - 2\delta_k$,

$$W_1(\tilde{\mathcal{L}}_{k+1}, \mathcal{L}^\star) \leq d W_1(\mathcal{L}_k, \mathcal{L}^\star) + \frac{cd_2\mathrm{diam}(\mathcal{A})\sqrt{|\mathcal{A}|}}{2}\epsilon_k + d_2\mathrm{diam}(\mathcal{A})|\mathcal{A}|e^{-c\phi(\epsilon)}. \qquad (30)$$

Finally, it is clear that with probability at least $1 - 2\delta_k$,

$$\begin{aligned}
W_1(\mathcal{L}_{k+1}, \mathcal{L}^\star) &\leq W_1(\tilde{\mathcal{L}}_{k+1}, \mathcal{L}^\star) + W_1(\tilde{\mathcal{L}}_{k+1}, \mathbf{Proj}_{S_\epsilon}(\tilde{\mathcal{L}}_{k+1})) \\
&\leq d W_1(\mathcal{L}_k, \mathcal{L}^\star) + \frac{cd_2\mathrm{diam}(\mathcal{A})\sqrt{|\mathcal{A}|}}{2}\epsilon_k + d_2\mathrm{diam}(\mathcal{A})|\mathcal{A}|e^{-c\phi(\epsilon)} + \epsilon.
\end{aligned}$$

By telescoping, this implies that with probability at least $1 - 2\sum_{k=0}^{K-1}\delta_k$,

$$\begin{aligned}
W_1(\mathcal{L}_K, \mathcal{L}^\star) &\leq d^K W_1(\mathcal{L}_0, \mathcal{L}^\star) + \frac{cd_2\mathrm{diam}(\mathcal{A})\sqrt{|\mathcal{A}|}}{2}\sum_{k=0}^{K-1} d^{K-k}\epsilon_k \\
&\qquad + \frac{(d_2\mathrm{diam}(\mathcal{A})|\mathcal{A}|e^{-c\phi(\epsilon)} + \epsilon)(1 - d^K)}{1 - d}.
\end{aligned} \qquad (31)$$

Since $\epsilon_k$ is summable, hence $\sup_{k\geq 0}\epsilon_k < \infty$, $\sum_{k=0}^{K-1}d^{K-k}\epsilon_k \leq \frac{\sup_{k\geq 0}\epsilon_k}{1-d}d^{\lfloor(K-1)/2\rfloor} + \sum_{k=\lceil(K-1)/2\rceil}^{\infty}\epsilon_k$.

Now plugging in $K = K_{\epsilon,\eta}$, with the choice of $\delta_k$ and $c = \frac{\log(1/\epsilon)}{\phi(\epsilon)}$, and noticing that $d \in [0,1)$, it is clear that with probability at least $1 - 2\delta$,

$$
\begin{aligned}
W_1(\mathcal{L}_{K_{\epsilon,\eta}}, \mathcal{L}^\star) \leq\ & d^{K_{\epsilon,\eta}}W_1(\mathcal{L}_0, \mathcal{L}^\star) \\
& + \frac{cd_2\mathrm{diam}(\mathcal{A})\sqrt{|\mathcal{A}|}}{2}\left(\frac{\sup_{k\geq 0}\epsilon_k}{1-d}d^{\lfloor(K_{\epsilon,\eta}-1)/2\rfloor} + \sum_{k=\lceil(K_{\epsilon,\eta}-1)/2\rceil}^{\infty}\epsilon_k\right) \\
& + \frac{(d_2\mathrm{diam}(\mathcal{A})|\mathcal{A}|+1)\epsilon}{1-d}.
\end{aligned}
\tag{32}
$$

Setting $\epsilon_k = (k+1)^{-(1+\eta)}$, then when $K_{\epsilon,\eta} \geq 2(\log_d(\epsilon/c)+1)$,

$$
\frac{\sup_{k\geq 0}\epsilon_k}{1-d}d^{\lfloor(K_{\epsilon,\eta}-1)/2\rfloor} \leq \frac{\epsilon/c}{1-d}.
$$

Similarly, when $K_{\epsilon,\eta} \geq 2(\eta\epsilon/c)^{-1/\eta}$, $\sum_{k=\lceil\frac{K_{\epsilon,\eta}-1}{2}\rceil}^{\infty}\epsilon_k \leq \epsilon/c$.

Finally, when $K_{\epsilon,\eta} \geq \log_d(\epsilon/(\mathrm{diam}(\mathcal{S})\mathrm{diam}(\mathcal{A})))$, $d^{K_{\epsilon,\eta}}W_1(\mathcal{L}_0, \mathcal{L}^\star) \leq \epsilon$, since $W_1(\mathcal{L}_0, \mathcal{L}^\star) \leq \mathrm{diam}(\mathcal{S}\times\mathcal{A}) = \mathrm{diam}(\mathcal{S})\mathrm{diam}(\mathcal{A})$.

In summary, if $K_{\epsilon,\eta} = \lceil 2\max\{(\eta\epsilon/c)^{-1/\eta}, \log_d(\epsilon/\max\{\mathrm{diam}(\mathcal{S})\mathrm{diam}(\mathcal{A}), c\})+1\}\rceil$, then with probability at least $1 - 2\delta$,

$$
W_1(\mathcal{L}_{K_{\epsilon,\eta}}, \mathcal{L}^\star) \leq \left(1 + \frac{d_2\mathrm{diam}(\mathcal{A})\sqrt{|\mathcal{A}|}(2-d)}{2(1-d)} + \frac{(d_2\mathrm{diam}(\mathcal{A})|\mathcal{A}|+1)}{1-d}\right)\epsilon = O(\epsilon).
$$

Finally, plugging in $\epsilon_k$ and $\delta_k$ into $T^{\mathcal{M}_L}(\delta_k, \epsilon_k)$, and noticing that $k\leq K_{\epsilon,\eta}$ and $\sum_{k=0}^{K_{\epsilon,\eta}-1}(k+1)^\alpha \leq \frac{K_{\epsilon,\eta}^{\alpha+1}}{\alpha+1}$, we immediately arrive at

$$
T = O\left((\log(K_{\epsilon,\eta}/\delta))^{\frac{1}{1-h}}K_{\epsilon,\eta}(\log K_{\epsilon,\eta})^{\frac{1}{1-h}} + (\log(K_{\epsilon,\eta}/\delta))^{\frac{1}{h}+3}\frac{K_{\epsilon,\eta}^{1+\frac{2(1+\eta)}{h}}}{1+\frac{2(1+\eta)}{h}}(\log(K_{\epsilon,\eta}/\delta))^{\frac{1}{h}}\right).
$$

By further relaxing $\eta$ to 1 and merging the terms, (11) follows. $\qquad\square$

## F   Naive algorithm

The Naive iterative algorithm (Algorithm 2) is to replace Step A in the three-step fixed-point approach of GMFGs with Q-learning iterations. The limitation of this Naive algorithm has been discussed in the main text (Step 1, Section 4) and empirically verified in Section 5 (Figure 4).

---

**Algorithm 2** Alternating Q-learning for GMFGs (Naive)

---

1: **Input**: Initial population state-action pair $L_0$
2: **for** $k = 0, 1, \cdots$ **do**
3:    Perform Q-learning to find the Q-function $Q_k^\star(s,a) = Q_{L_k}^\star(s,a)$ of an MDP with dynamics $P_{L_k}(s'|s,a)$ and rewards $r_{L_k}(s,a)$.
4:    Solve $\pi_k \in \Pi$ with $\pi_k(s) = \textbf{argmax-e}\,(Q_k^\star(s,\cdot))$.
5:    Sample $s \sim \mu_k$, where $\mu_k$ is the population state marginal of $L_k$, and obtain $L_{k+1}$ from $\mathcal{G}(s, \pi_k, L_k)$.
6: **end for**

---

# G GMF-V

GMF-V, briefly mentioned in Section 4, is the value-iteration version of our main algorithm GMF-Q. GMF-V applies to the GMFG setting with fully known transition dynamics $P$ and rewards $r$.

---

**Algorithm 3 Value Iteration for GMFGs (GMF-V)**

---

1: **Input**: Initial $L_0$, tolerance $\epsilon > 0$.
2: **for** $k = 0, 1, \cdots$ **do**
3:      Perform value iteration for $T_k$ iterations to find the approximate Q-function $Q_{L_k}$ and value function $V_{L_k}$:
4:      **for** $t = 1, 2, \cdots, T_k$ **do**
5:        **for** all $s \in \mathcal{S}$ and $s \in \mathcal{A}$ **do**
6:          $Q_{L_k}(s, a) \leftarrow \mathbb{E}[r(s, a, L_k)] + \gamma \sum_{s'} P(s'|s, a, L_k) V_{L_k}(s')$
7:          $V_{L_k}(s) \leftarrow \max_a Q_{L_k}(s, a)$
8:        **end for**
9:      **end for**
10:      Compute a policy $\pi_k \in \Pi$:
           $\pi_k(s) = \mathbf{softmax}_c(Q_{L_k}(s, \cdot))$.
11:      Sample $s \sim \mu_k$, where $\mu_k$ is the population state marginal of $L_k$, and obtain $\tilde{L}_{k+1}$ from $\mathcal{G}(s, \pi_k, L_k)$.
12:      Find $L_{k+1} = \mathbf{Proj}_{S_\epsilon}(\tilde{L}_{k+1})$
13: **end for**

---

# H More details for the experiments

## H.1 Competition intensity index $M$.

In the experiment, the competition index $M$ is interpreted and implemented as the number of selected players in each auction competition. That is, in each round, $M - 1$ players will be randomly selected from the population to compete with the *representative* advertiser for the auction. Therefore, the population distribution $\mathcal{L}_t$, the winner indicator $w_t^M$, and second-best price $a_t^M$ all depend on $M$. This parameter $M$ is also referred to as the *auction thickness* in the auction literature [18].

## H.2 Adjustment for Algorithm MF-Q.

For MF-Q, [40] assumes all $N$ players have a joint state $s$. In the auction experiment, we make the following adjustment for MF-Q for computational efficiency and model comparability: each player $i$ makes decision based on her own private state and table $Q^i$ is a functional of $s^i$, $a^i$ and $\frac{\sum_{j \neq i} a^j}{N-1}$.