[Reviews · NeurIPS 2019]

Reviewer 1



The framework proposed in this paper addresses the problem of multi-agent online decision-making when the population size is large. The problem is well motivated by practical examples such as online ad auctions. The results stated here are nice. Some references to background knowledge should be provided: 1. In Section 2.1, it is claimed that N-player game is hard, but this needs some verification. 2. In Section 3, what is the definition of Wasserstein distance? There also exist some confusions in notations and possible typos: 1. In line 67, what is s^t? It seems like the authors are referring to s_t. 2. In line 78, the definition of the indicator function is unclear. It would indeed be good if the authors can use a separate section to state notations.

Reviewer 2



This paper considers learning in mean-field games (MFG). MFGs take the limit of an infinite number of agents, which are considered indistinguishable. Based on a motivating example consisting of a repeated Ad auction problem, the authors introduce a "general" mean-field game (GMFG), a model-free version of the standard MFG. The authors revisit standard Q-Learning and a soft version of it, and provide convergence and complexity results of such an algorithm. These methods are compared numerically on the auction problem together with a recently proposed approach and show better performance. The paper addresses a relevant problem in mean field games, a body of work that is becoming increasingly popular. It is overall well written and provides interesting results. I have the following concerns with this paper: - stationary vs infinite-time policies: it is unclear what are the assumptions required for the stationary case. The original formulation [15] considers the finite-horizon (time-dependent policies) setting. The authors just remove the time index and go with that. This should be clarified. - Incremental contribution: Looking at [12,18], the theoretical contribution seems too incremental. The main difference being the use of or state-action distributions (to define Q values) instead of the state distribution only. Regarding this, I think the term General Mean Field Games (GMFG) is confusing, since the presented is just a different (model-free way) of solving the same class of Mean Field games. It is unclear in what way GMFGs are more general than, e.g., in [25]. - Soft version of Q-learning: the literature on complexity/convergence of the soft version of Q-learning used in the paper is marginally cited. For example, the results of Asadi & Littman "An Alternative Softmax Operator for Reinforcement Learning" which shows potential divergence when using the Botzmann operator, as seems to be used in this paper. - Generality of the repeated Ad auction problem: the method is only illustrated on the auction problem, which is interesting, but limited. It reduces winning an auction to the game intensity parameter M. This seems to simplify the problem significantly. It would be desirable to see how the method performs in other popular benchmarks for MFGs. - Related work on MFGs: The authors are aware of the very recent AAMAS paper [25], but avoid to relate both works. There are other relevant papers that should also be discussed: * Mean field stochastic games with binary actions: Stationary threshold policies. M. Huang and Y. Ma. CDC 2017 * Decentralised Learning in Systems With Many, Many Strategic Agents. D, Mguni et al. AAAI 2018. minor: I found confusing how the notion of Nash equilibrium used in this paper. Since NE involves a optimization between different agents, it is clear that an individual agent is optimizing the cost given the joint distribution. However, the "population side" is just inferring the "compatible" joint with the individual optimal policy. line 121: a^M should be a^M_t line 152 (and 419): P(\cdot|\cdot, ...) should be P(\cdot|s_t, ...) line 211: use a different symbol for \alpha, since \alpha is used for the marginal before Algorithm 1: line 1: where is parameter c? line 3: better don't mention the model, since it is a model-free step line 251: "iteration" -> iterations line 261: "It learning accuracy" -> "Its learning accuracy" line 408: remove subindex $t$ of $\pi_t$ in $\pi_t(s_t,\mathcal{L})$ ----------------------- POST REBUTTAL After the authors rebuttal and the discussion I lean towards acceptance. However, I think the paper needs substantial editing to clarify the setting, specially stationary vs non-stationary, finite-horizon. It should be better clarified how the time-independent solutions found by the proposed algorithm are captured in a non-stationary setting. The paper also lacks an in-depth comparison with papers [31] and [15], describing differences in the settings, e.g., function approximation, stationarity and the obtained results.

Reviewer 3



1. This work seems to be the first to propose a convergent RL algorithm for MFG that is able to approximately find the mean-field equilibrium. Moreover, the explanation of why the naive application of Q=learning fails seems convincing. 2. The paper is well-written overall. However, the presentation can be further improved. For examples, in Section 2.3, it would be better to directly state the problem of the repeated auction as an MFG by specifying what $L$ is and how the reward function and transition probability depends on the mean-field distribution. Currently, the reward function in (3) seems to have a lot of notations and it is unclear how to write it as a function of s, a, and L. Another example is in Theorem 2, it would be better to carefully define each quantity instead of deferring them to the appendix. 3. In Section 4 the authors focus on the stationary solution which is time-independent. However, the uniqueness and existence theorem only guarantee a solution which might depend on time. Is it possible to directly show that the MFG solution is time-independent? 4. Given L, since the optimal policy of an MDP can be non-unique. In this case, the left-hand side of (5) might be large. Is there any intuition why this assumption holds? 5. My major concern is on the epsilon-net. Since the space of all distribution over S\times A is large, the size of this epsilon-net might be huge. In practice, how to construct this epsilon-net? In addition, the construction of this epsilon-net yields a gap $phi(\epsilon)$, which appears in the parameter $c$. In practice, how to choose parameter $c$? 6. The number of iteration in Theorem 12 seems complicated. Is there an easy way to see the order of iteration complexity? 7. Existence of mean-field equilibrium in discrete-time finite MFG is studied in the literature. See Markov--Nash Equilibria in Mean-Field Games with Discounted Cost“

[Author Response · NeurIPS 2019]

We thank all three reviewers for their careful reading and constructive suggestions. We will revise the paper thoroughly,
incorporating all the comments.

**[reviewer 1]** We will provide precise references from the classical literature on the hardness of $N$-player games,
including [PR05]. We will add the definition of the $\ell_1$-Wasserstein distance to make the paper self-contained. In
addition, we will correct $s^t$ (of line 67) to $s_t$, and rewrite the formula $\mu_t(s)$ as $\mu_t(s) = \frac{\sum_{j=1, j \neq i}^{N} 1_s(s_t^j)}{N}$, where the
indicator function $1_s(s_t^j) = 1$ if $s_t^j = s$ and 0 otherwise. Finally, we will add a section to clearly define all notations.

**[reviewer 2]** (1) We will revise the presentation carefully, as suggested. (2) Thank you for asking the clarification
between the stationary versus non-stationary MFGs. Stationary solutions are commonly adopted for MFGs with an
infinite-time horizon, see [12,15]. Non-stationary solutions are mostly used for MFGs with a finite-time horizon, see
[Bar12]. Our work shows the existence and the uniqueness theorems for both the stationary (Appendix B) and the
non-stationary MFGs (Theorem 1). The algorithms in Section 4 are focused on Q-learning algorithms, which are
primarily designed for stationary MDPs and hence appropriate for stationary MFGs.

(3) On the contribution: The GMFG framework (Section 3) incorporated *both* the state distribution and the action
distribution. With the additional action distribution, $\Gamma_2$ was different from the one defined in [12] and the proof for the
uniqueness and the existence of the solution needed further modifications. To clarify the difference with [25]: [25]
showed the convergence to a *local* (Nash equilibrium) solution, and the uniqueness of the local solution given the
presence of a unique *global* solution. However, [25] did not analyze the existence of a unique *global* solution. We
established the existence, the uniqueness, and the convergence to a *global* solution. We will add this discussion in the
revision.

(4) On the related works of MFG: Apologies for missing some references, which we will add with careful discussions
of their contributions and relationship to our work ([HM17, MJdC18]).

(5) On the literature related to soft Q-learning: We will include additional references. Thank you for pointing out the
potential divergence of SARSA using the Boltzmann operator [AL17]. Indeed, it is now more interesting to see the
guaranteed convergence with Q-learning using the Boltzmann policies. We will add this comparison and discussion in
the revision. Indeed, we think it is worth testing the performance using the Mellowmax exploration, in addition to the
Boltzmann exploration.

(6) On the definition of NE: MFG is a game with an infinite number of identical players. The NE solution is therefore
the same for each individual by symmetry. If each individual in the population follows the conditional optimal solution
(from the single player side), the consistency means that no player in the population has the incentive to deviate (from
the solution of the single player side). This is consistent with the NE definition for $N$-player games.

(7) For the Ad auction example, apologies for the confusion. $M$ is only one of several model parameters and indicates
the competition intensity. The game interaction is more extensive than $M$ alone: for each agent, all of her reward, her
winning probability and her budget dynamics, depend on the strategies from other opponents.

**[reviewer 3]** We will rewrite the repeated auction example in Section 2.3, in order to be consistent with the general
model setting in Section 2.2. We will also clearly define and explain the quantities in Theorem 2.

(3) For the stationary setting in Section 4, the corresponding uniqueness and existence theorems for the time-independent
MFG solution (i.e., Theorem 4) are given in Appendix B (see Line 174) under slightly different conditions from the
non-stationary setting. Note that due to the introduction of the mean information process in the MFG, an infinite-time
horizon MFG is generally associated with a parabolic type PDE, hence the Nash equilibrium could still be time
dependent. This is fundamentally different from the theory of single-agent MDP where the optimal control, if exists
uniquely, would be time independent in an infinite-time horizon setting.

(4) For Assumption 1 and inequality (5), we can impose $\Gamma_1$ to be single-valued by using *e.g.*, **argmax-e**. Moreover,
in the linear-quadratic continuous state-action setting, the assumption can be translated into constraints on model
parameters. We will add this in the revision.

(5) In practice, a uniform grid for the epsilon-net would suffice, as shown in our experiments. That is to replace the
projection of $\tilde{\mathcal{L}}_k$ onto the epsilon-net by truncating the resulting $\tilde{\mathcal{L}}_k$, up to a certain number of digits. For example, 4
was used in the experiment. The choice of $c$ is fairly simple, as the experiments are robust with respect to different
values of $c$, ranging from 1 to 100. For instance, we chose $c = 5$.

(6) For the iteration complexity in Theorem 2, indeed, one could simplify the order of iteration complexity, by simply
taking $h$ to be 3/4 and $\eta$ to be 1. We will clarify this.

(7) Thank you for the reference [SBR18] for discrete-time MFGs, which will be added accordingly.

# References

[AL17] K. Asadi and M. L. Littman. An alternative softmax operator for reinforcement learning. 2017.
[Bar12] M. Bardi. Explicit solutions of some linear-quadratic mean field games. 2012.
[HM17] M. Huang and Y. Ma. Mean field stochastic games with binary actions: Stationary threshold policies. 2017.
[MJdC18] D. Mguni, J. Jennings, and E. M. de Cote. Decentralised learning in systems with many, many strategic agents. 2018.
[PR05] C. H. Papadimitriou and T. Roughgarden. Computing equilibria in multi-player games. 2005.
[SBR18] N. Saldi, T. Basar, and M. Raginsky. Markov-Nash equilibria in mean-field games with discounted cost. 2018.


[Meta-Review · NeurIPS 2019]

All reviewers agree on the contribution of the work in establishing theoretical conditions for existence and uniqueness of NE in MGFs with unknown rewards and dynamics and proposing a soft Q-learning based algorithm and provide convergence analysis. Reviewer #2 had some concerns regarding the difference between stationary and non-stationary settings as well as significance in difference and missing discussions to prior related works. Most of these concerns are addressed in the rebuttal and reviewer discussion. We thus decide to accept the paper. Please incorporate reviewers' comments in preparing the camera-ready version.